# FINE-GRAINED PROMPT-DRIVEN STYLIZATION WITH CONTEXT-AWARE REASONING FOR ZERO-SHOT DOMAIN ADAPTATION

## ABSTRACT

Zero-shot domain adaptive semantic segmentation (ZSDA) aims to generalize models to unseen target domains without accessing target data during training. Recent methods commonly use vision-language models (VLMs) to simulate target-domain features by guiding stylization with textual prompts. However, these approaches often suffer from two key issues: *description mismatch*, where generic prompts fail to reflect scene-specific semantics, and *prompt-induced discrepancy*, where normalization guided by coarse prompts cannot capture spatial variations. Together, these problems lead to a noticeable *simulated-vs-real feature gap*, reducing adaptation effectiveness. To address this, we propose **FineDA**, a framework designed to reduce this gap through image-specific prompt reasoning and fine-grained feature stylization. FineDA introduces a *scene graph-guided chain-of-thought* module that generates contextual, semantically rich target descriptions for each source image. It also incorporates a *prompt-guided local and global stylization* module, enabling patch-wise class-specific adaptation while maintaining scene-level consistency. Extensive experiments on standard ZSDA benchmarks and a challenging in-house surgical dataset with adverse visual conditions such as smoke, blood, and low lighting demonstrate the effectiveness and generalization capability of our approach. Code will be released upon publication.

## 1 INTRODUCTION

Semantic segmentation is a fundamental task in computer vision, with broad applications in safety-critical scenarios such as surgical robotics, autonomous driving, and medical imaging(Hesamian et al., 2019; Hong et al., 2018; Siam et al., 2017). The success of recent segmentation models largely relies on the assumption that the training and testing data are drawn from the same distribution. However, this assumption is often violated due to variations in scene, sensor, or environmental conditions, leading to significant performance degradation. To mitigate this issue, domain adaptation (DA) methods have been proposed to transfer knowledge from a labeled source domain to an unlabeled target domain using various alignment strategies(Long et al., 2018; Murez et al., 2018; Li et al., 2019). When target domain data is not available during training, domain generalization (DG) methods aim to learn representations that generalize to unseen domains (Li et al., 2018a;b; Wang et al., 2022).

Recent advances in vision-language models (VLMs), such as CLIP(Radford et al., 2021), have inspired new approaches for zero-shot domain adaptive (ZSDA) semantic segmentation. These methods leverage the strong generalization ability of VLMs by injecting domain-specific textual prompts into the model to simulate target-domain styles. A common paradigm is a two-stage pipeline, which is to augment source-domain images using textual guidance that reflects target conditions, and then use these augmented features to train segmentation models. For example, PØDA(Fahes et al., 2022) proposed Prompt-driven Instance Normalization (PIN) with coarse target descriptions such as "car driving in rain". While it provide substantial improvements for ZSDA, achieving a satisfactory alignment with the target domain is particularly challenging for the model when it relies solely on adapting the global feature to fit the target domain. To alleviate this issue, ULDA (Yang et al., 2024) achieving complex alignment of features at multiple levels, including scene-level, region-level, and pixel-level.

While recent vision-language approaches have shown promising results for zero-shot domain adaptive segmentation, they rely on two strong assumptions that often do not hold in practice, as illustrated in Figure 1. First, they assume that a coarse textual prompt can accurately describe the semantics and style of the target image. However, such prompts frequently suffer from **description mismatch**, failing to capture scene-specific structures and object relationships. Second, they presume that features stylized through these prompts can effectively represent the target domain. In reality, this mismatch, combined with **prompt-induced discrepancy** from applying a uniform global transformation, leads to a **simulated versus real feature gap**, where adapted features deviate from true target characteristics.

To address these issues, we propose *FineDA*, a framework that combines image-aware prompt reasoning with fine-grained stylization to reduce the gap between simulated and real target features. Firstly, inspired by Mitra et al. (2023), FineDA introduces a Scene Graph-guided Chain-of-Thought module that generates image-specific, semantically structured descriptions by extracting object attributes and relationships, ensuring spatial and relational semantics are preserved. These contextualized prompts then guide a Prompt-driven Local and Global Feature Stylization module, which extends prompt-based instance normalization (Fahes et al., 2022) to perform patch-wise class-specific and global normalization, enabling precise and adaptive simulation of domain shifts. In the second stage, stylized features are used to train a segmentation model with a Visual-text Feature Alignment Regularizer, following the DenseCLIP (Rao et al., 2022) formulation, which aligns semantic features and class-specific text embeddings in CLIP space. Experiments on standard ZSDA benchmarks and a challenging surgical dataset with smoke and blood disturbances validate the effectiveness and robustness of our approach.

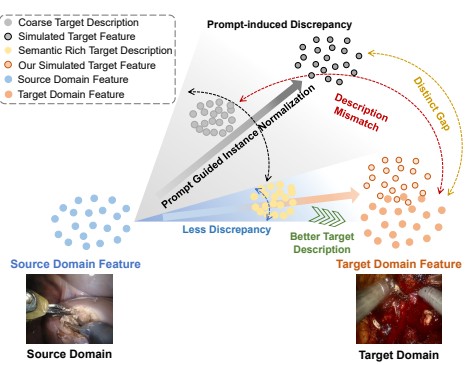

Figure 1: Previous prompt-guided stylization methods often suffer from prompt-induced discrepancy during normalization and mismatched descriptions that ignore image-specific semantics. Together, these issues lead to a *simulated-vs.-real gap*, where the adapted source features fail to accurately reflect target-domain representations. Our method addresses these issues through image-aware Chain-of-Thought descriptions and joint local-global stylization for fine-grained reliable feature adaptation.

Our main contributions are summarized as follows:

- We present a novel framework for zero-shot domain adaptive semantic segmentation that improves prompt quality for better target domain description, and enables fine-grained feature adaptation to spatially non-uniform domain shifts.
- We develop a Scene Graph-guided CoT module to generate image-specific, semantically rich target descriptions, and a Prompt-driven Local and Global Feature Stylization module that performs patch-wise class-aware adaptation guided by both local and global prompts.
- We design a Visual-text Feature Alignment Regularizer based on CLIP-style supervision to enhance segmentation learning from stylized features. Extensive experiments on standard benchmarks and a challenging surgical dataset demonstrate the effectiveness and robustness of our approach.

## 2 RELATED WORK

**Unsupervised Domain Adaptation.** UDA has been widely explored to improve model generalization to unseen domains. A prevalent strategy is pseudo-label self-training, where pseudo labels are generated by either a teacher mode(Chen et al., 2023; Gong et al., 2023) or the model itself(Kothandaraman et al., 2021; Zou et al., 2018). To improve label quality, several works focus on refining or stabilizing these pseudo labels(Yao & Li, 2023; Zhang et al., 2023a). Other approaches include adversarial domain alignment Gong et al. (2021), contrastive representation learning(Cheng et al., 2023; Tsai et al., 2023), regularization-based techniques(Chen et al., 2018b; Huo et al., 2022), multi-resolution feature modeling(Ma et al., 2022; Peng et al., 2018), and architecture-aware adaptation(Mascaro et al., 2023; Zhang et al., 2023b). However, in real-world applications such as indus-

trial or medical settings(Ma et al., 2022), access to representative target-domain data is often limited or infeasible, motivating research in target-free or zero-shot domain adaptation.

**Zero-Shot Domain Adaptation.** Zero-shot domain adaptation operates under more stringent conditions compared to the conventional scenario, where no target-domain images are available. Some studies (Lengyel et al., 2021; Luo et al., 2023) demonstrated zero-shot day-night adaptation, where no target images were used but with a specific limitation to the day-night scenario. For a more general setting, a few recent works (Fahes et al., 2022; Vidit et al., 2023; Yang et al., 2024; Sikdar et al., 2025) exploit CLIP (Radford et al., 2021), a vision-language pre-trained model. These approaches leverage text descriptions of the target domain, i.e., prompts, via the PIN module to adapt a source-trained model to target domains. Nevertheless, such approaches use fixed target domain text descriptions as prompts to guide feature enhancement, which fails to account for the contextual information inherent to each individual image and creates a bias between simulated and real target features. Another recent line of work explores synthetic data generation (Luo et al., 2025; Azuma et al., 2024; Kim et al., 2025), where stable diffusion is employed to produce target-style images for adaptation. While effective on large-scale benchmarks, their performance is highly dependent on the quality of the generative model. In complex domains such as surgical images, where visual disturbances are common, the generated results are often unstable, limiting their effectiveness. PIN approaches, in contrast, offer a lightweight alternative, and the two paradigms are not mutually exclusive, as demonstrated by SIDA (Kim et al., 2025), which integrates instance normalization with diffusion-generated data. In this work, we propose to bridge the gap by generating context-aware descriptions tailored to each source image and performing patch-wise class-specific normalization. This design reduces the bias between simulated and real target features while maintaining semantic fidelity under diverse and non-uniform domain shifts. Our framework is also complementary to diffusion-based strategies like SDGPA (Luo et al., 2025), as context-aware reasoning and fine-grained normalization can be integrated with synthetic data generation to further improve zero-shot adaptation.

## 3 METHODOLOGY

Our framework improves zero-shot domain adaptive semantic segmentation by integrating contextual prompt generation with fine-grained feature adaptation as shown in Figure 2. Given a source image, we first generate a target-style textual description through contextual reasoning based on a scene graph and a large language model, enabling the prompt to better align with the real target domain. Guided by these context-rich descriptions, we introduce a prompt-driven local and global feature stylization module that simulates target-domain styles in a class-aware and region-sensitive manner. The adapted features are then fed into a segmentation head, where a visual-text feature alignment regularizer is utilized to compute per-pixel similarity between image features and CLIP-derived class embeddings. This enables effective segmentation without requiring any access to target-domain images.

### 3.1 PRELIMINARY

To incorporate the textual description of target domain, PØDA (Fahes et al., 2023) proposed a two-stage ZSDA framework. It relies solely on a general text description of the target domain, thereby removing the need for target domain images during training. The method consists of two main stages: i) the stylization of features from the target domain, and ii) the fine-tuning process of the target segmenter.

**Stage-1:** Inspired by AdaIN(Huang & Belongie, 2017), PØDA introduced a Prompt-driven Instance Normalization (PIN) module to transform the source image features $f_s$ extracted from the CLIP image encoder to the target stylized features $f_{s \to t}$. The transformation is guided by the alignment between $f_{s \to t}$ and the target description given as a text prompt, such as "car in the rain". The stylization process is mathematically formulated as follows:

$$f_{s \to t} = \text{PIN}(f_s, \mu, \sigma) = \sigma \left( \frac{f_s - \mu(f_s)}{\sigma(f_s)} \right) + \mu. \tag{1}$$

where $\mu$ and $\sigma$ are trainable parameters that represent the style information of target domain features. $\mu(f_s)$ and $\sigma(f_s)$ represent the mean and standard deviation of the source input features $f_s$.

This operation is followed by an attention-based pooling operation, resulting in the output denoted as $\bar{f}_{s\to t}$. By applying the loss function presented in Equation (2), the PIN module improves the similarity between $f_{s\to t}$ and the CLIP text embedding TrgEmb, which characterizes the style of the target domain.

$$\mathcal{L}_{\mu,\sigma}\left(\bar{f}_{s\to t}, \text{TrgEmb}\right) = 1 - \frac{\bar{f}_{s\to t} \cdot \text{TrgEmb}}{\left\|\bar{f}_{s\to t}\right\| \left\|\text{TrgEmb}\right\|} \tag{2}$$

**Stage-2:** After training the PIN module using source images and their corresponding target domain descriptions, we can transform each source image feature into its target domain counterpart. The target segmenter can be obtained by fine-tuning the classification head with these simulated features and their labels.

### 3.2 SCENE GRAPH-GUIDED CHAIN-OF-THOUGHT FOR TARGET DESCRIPTION GENERATION

To generate a more detailed and target-specific description that better guides the style mining process, we adopt a two-step chain-of-thought reasoning approach. As illustrated in Figure 2, the first step involves generating a scene graph $S_g$. The input prompt for this step, denoted as $P_{\text{in}}^1$, is composed of the input image $I$, a domain-specific prompt $P_t$, and a scene graph instruction prompt $P_s$.

The scene graph prompt $P_s$ is designed to instruct the vision-language model (VLM) to construct a structured scene representation with three core components: (1) **objects**, which are extracted from the ground truth segmentation mask; (2) **attributes** of these objects; and (3) **relationships** between them. This structured representation ensures that contextual and semantic information is preserved throughout the subsequent reasoning process. The complete prompt formulation is given in Eq. equation 3. Based on this input, the VLM generates the corresponding scene graph $S_g$.

$$P_{\text{in}}^1 = \text{``}[I][P_t]\,[P_s]\text{''} \tag{3}$$

In the second step, the VLM is prompted with the input image $I$, the domain prompt $P_t$, the generated scene graph $S_g$, and a reasoning instruction prompt $P$. The prompt $P$ guides the VLM to utilize the provided context effectively, and is defined as: "*Use the image and scene graph as context, and reason about the task.*" Based on this comprehensive input, the VLM generates a target-domain description for the source image. All four components are jointly used as contextual information to support the reasoning process. This enriched context encourages the model to produce target-domain descriptions that are both semantically aligned and domain-specific. The overall input prompt for this response generation step is formulated as:

$$P_{\text{in}}^2 = \text{``}[I]\,[P_t]\,[S_g]\,[P]\text{''} \tag{4}$$

To ensure compatibility with CLIP, we use an LLM to simplify the VLM-generated descriptions by filtering out irrelevant content based on the ground truth mask, preserving essential context.

### 3.3 PROMPT-DRIVEN LOCAL AND GLOBAL FEATURE STYLIZATION

We propose the Prompt-driven Global and Local Feature Stylization (PLGFS) module to adapt source features more effectively under complex and spatially non-uniform domain shifts. The motivation behind PLGFS is twofold. First, PLGFS builds upon the detailed and target-specific descriptions generated through a two-step chain-of-thought reasoning process in the previous stage. These descriptions provide rich semantic context that enhances the style mining process, allowing the model to better align source features with target-domain semantics. Second, domain shifts in real-world scenarios (e.g., surgical scenes) often exhibit localized variations due to lighting, occlusions, or fluids, making global alignment insufficient. To address this, we adopt a patch-wise normalization strategy driven by class-specific prompts. This allows local features to be individually stylized according to their semantic content, while maintaining global coherence through image-level prompts. By integrating both global and local prompt guidance, PLGFS performs targeted instance normalization that improves semantic alignment and robustness for downstream domain adaptation.

**Local prompt-driven instance normalization.** To enable fine-grained adaptation, we construct patch-level visual-textual pairs and apply localized prompt-driven stylization. Given a batch of

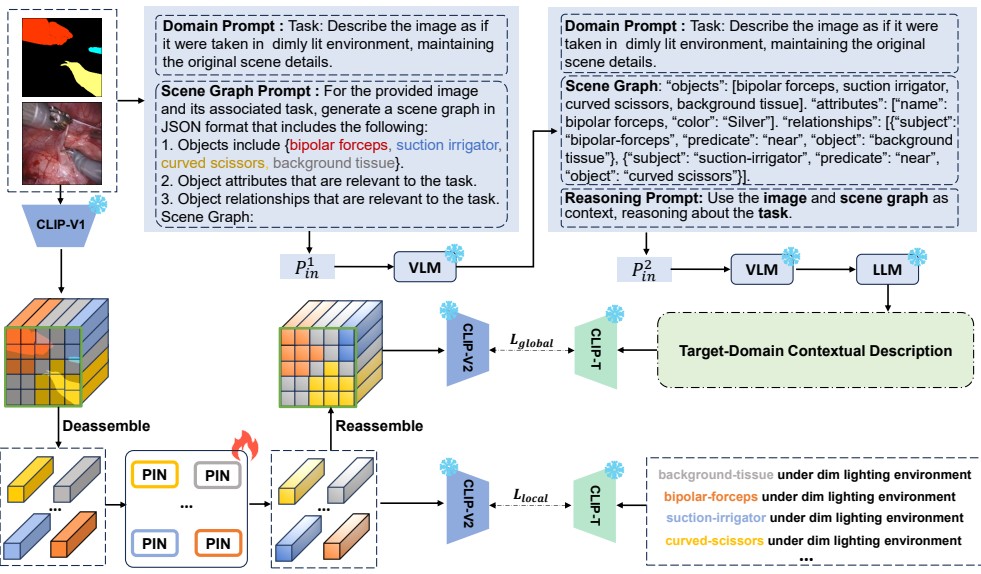

Figure 2: Stylization process in the FineDA framework. A scene graph is constructed from the source image and segmentation mask to generate an image-specific target description via Chain-of-Thought reasoning. Patch-wise features are then locally stylized using class-specific prompts, and refined globally with the generated description to ensure consistent and domain-aligned adaptation. CLIP-V1 is the stem layers and Layer 1 of the CLIP image encoder, CLIP-V2 is the remaining layers excluding the attention pooling.

source images $I_s$, we extract low-level features $\mathcal{F}_s$ using the first layer of the CLIP image encoder CLIP-V1. Each feature map $f_s \in \mathcal{F}_s$ is divided into $m$ non-overlapping patches, yielding a set of patch features $f_p \in \mathbb{R}^{\frac{h}{\sqrt{m}} \times \frac{w}{\sqrt{m}} \times c}$. To assign semantics to each patch, we determine its dominant class using the corresponding region in the ground truth segmentation mask. For each patch label $y_p$, we compute the frequency of class labels and select the one with the highest occurrence as the dominant class $c_p$, which is then mapped to its textual name $t_p$ using a predefined vocabulary.

We then construct a local prompt $P_p$ by concatenating the class name $t_p$ with a sampled style phrase $R$ from the style prompt pool $\mathcal{R}$, forming phrases such as *"bipolar-forceps under dim lighting conditions."* Each pair $(f_p, P_p)$ is then used to guide localized feature stylization. We build upon the Prompt-Instance Normalization (PIN) method, which learns to modulate feature statistics based on prompts. Different from the original PIN that performs global normalization, we apply PIN in a *patch-wise* manner. Specifically, for each patch feature $f_p$, we use its corresponding prompt $P_p$ to predict the target mean and standard deviation. These statistics are then used to normalize the patch feature as:

$$\hat{f}_p = \sigma(P_p) \cdot \frac{f_p - \mu(f_p)}{\sigma(f_p)} + \mu(P_p), \tag{5}$$

where $\mu(f_p)$ and $\sigma(f_p)$ are the mean and standard deviation of the input patch feature, and $\mu(P_p)$, $\sigma(P_p)$ are learned target statistics predicted from the text prompt $P_p$. As a result of this process, we obtain $m$ predicted mean and standard deviation pairs, each corresponding to one spatial patch. These parameters encode class-aware and spatially adaptive style information, enabling the model to tailor feature distributions at a fine-grained level. This localized operation enhances the model's ability to handle non-uniform domain shifts and improves overall feature robustness in complex visual environments.

**Global prompt-driven refinement.** While local prompts provide class-specific guidance for each patch, they do not capture the broader semantic context of the entire image. To enhance global consistency, we introduce a refinement stage that incorporates global scene-level information.

To incorporate global scene-level consistency, we reassemble the $m$ patch features to construct a global image feature representation. We then align this global feature with the global context prompt $\mathcal{G}$, which is obtained from the chain-of-thought reasoning stage. This step enables the global feature

to align with target-domain semantics beyond the scope of individual patches. By integrating both local and global prompt-driven stylization, the final representation captures fine-grained variations while maintaining global semantic consistency. The resulting stylized features $\mathcal{S}_{s \to t}$ are then used for downstream domain adaptation tasks.

**Loss Function.** To supervise the prompt-driven stylization process, we adopt a contrastive learning objective that aligns visual features with their corresponding textual prompts in a shared embedding space.

For the local stage, we compute a contrastive loss between each patch feature $f_p$ and its associated local prompt $P_p$. Following the CLIP training paradigm, we encourage each visual-textual pair to have high similarity while pushing apart mismatched pairs. The local alignment loss is defined as:

$$\mathcal{L}_{\text{local}} = -\log \frac{\exp(\text{sim}(f_p, P_p)/\tau)}{\sum_{P' \in \mathcal{P}} \exp(\text{sim}(f_p, P')/\tau)}, \tag{6}$$

where $\text{sim}(\cdot)$ denotes cosine similarity between normalized embeddings, $\tau$ is a temperature parameter, and $\mathcal{P}$ is the set of all prompts in the batch.

For the global stage, after reassembling the patch features into a global feature $f_g$, we compute a similar contrastive loss with the global context prompt $\mathcal{G}$:

$$\mathcal{L}_{\text{global}} = -\log \frac{\exp(\text{sim}(f_g, \mathcal{G})/\tau)}{\sum_{\mathcal{G}' \in \mathcal{G}_{\text{batch}}} \exp(\text{sim}(f_g, \mathcal{G}')/\tau)}. \tag{7}$$

Finally, the total loss is defined as a weighted sum of the two components:

$$\mathcal{L}_{\text{total}} = \lambda_{\text{local}}\mathcal{L}_{\text{local}} + \lambda_{\text{global}}\mathcal{L}_{\text{global}}, \tag{8}$$

where $\lambda_{\text{local}}$ and $\lambda_{\text{global}}$ are hyperparameters controlling the contribution of each loss term.

### 3.4 VISUAL-TEXT FEATURE ALIGNMENT REGULARIZER

After the feature stylization procedure, we utilized the simulated target domain features to fine-tune the segmentation head, enabling the model to be effectively adapted to the target domain. To bridge this gap and enhance the model's adaptation to the target domain's finer details, we introduce an additional regularization term during the second stage. This term aims to refine the model's predictions at the pixel level, focusing on pixel-level semantic alignment between image features and target domain descriptions. To this end, Inspired by the DenseCLIP, which calculates a Cost Map by measuring the similarity between image features and textual embeddings, We adapt this approach to compute the similarity from features extracted from the layer preceding the prediction head, using it as an auxiliary regularization loss. Specifically, as shown in in Figure 3. Given an a set of classes C, by utilizing the C semantic classes and the target domain description, we can use CLIP text encoder to extract the text embedding $T \in \mathbb{R}^{C \times d}$. Then we extract features $V \in \mathbb{R}^{(H \times W) \times d}$ from the layer preceding the prediction head. We use the image and text embeddings $V(i)$ and $T(c)$, where $i$ denotes the 2D spatial positions of the image embedding and $c$ denotes an index for a class, to compute a cost map $M \in \mathbb{R}^{(H \times W) \times C}$ by cosine similarity. Formally, this is defined as

$$M(i, c) = \frac{V(i) \cdot T(c)}{\|V(i)\|\|T(c)\|} \tag{9}$$

The pixel-wise ground-truth annotation $y \in \mathbb{R}^{H \times W}$ can be accordingly transformed into $y' \in \mathbb{R}^{H \times W \times C}$, Each pixel $i$ has a label $y'(i, c)$ in one-hot encoding format. If pixel $i$ belongs to class $c$, then $y'(i, c) = 1$; otherwise, $y'(i, c) = 0$. We compute an auxiliary cross-entropy loss between the prediction $P$ and ground truth $y' \in \mathbb{R}^{H \times W}$, which serves as a regularization term for the segmentation head. We define this alignment loss as follows:

$$\mathcal{L}_{\text{cost}} = -\sum_{i=1}^{H \times W} \sum_{c=1}^{C} y(i, c) \log(M(i, c)) \tag{10}$$

Therefore, the overall loss function $\mathcal{L}_{\text{total}}$ becomes:

$$\mathcal{L}_{\text{total}} = \mathcal{L}_{\text{seg}} + \lambda \mathcal{L}_{\text{cost}} \tag{11}$$

where $\lambda$ is a weighting factor that controls the contribution of the auxiliary loss $\mathcal{L}_{\text{cost}}$ to the total loss function.

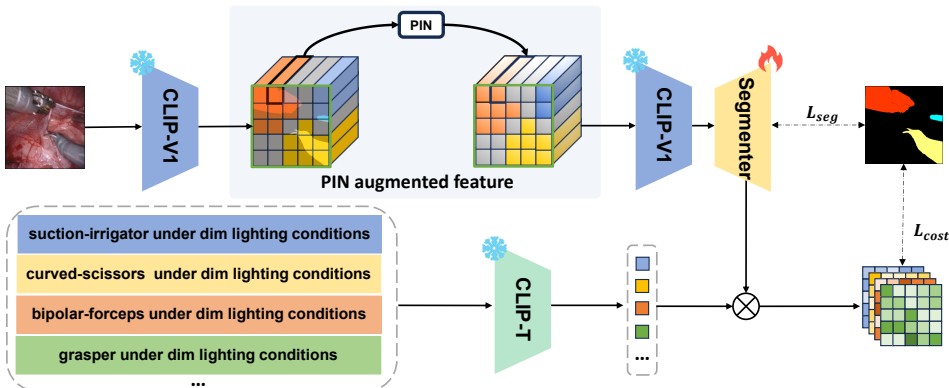

Figure 3: Training of segmentation model with Visual-Text Feature Alignment Regularizer.

# 4 EXPERIMENTAL RESULTS

## 4.1 EXPERIMENTAL SETUP

**Dataset and evaluations.** We primarily use the Cityscapes (Cordts et al., 2016) as the source domain dataset. Following PØDA, we report the main results tested on ACDC (Sakaridis et al., 2021) and GTA5 (Richter et al., 2016). To demonstrate the generalization of our method, we also investigate domain adaptation performance in surgical scenes, the source data is EndoVis2018 (Allan et al., 2020) and target domain image is an in-house surgical scence dataset SurgDVC. EndoVis2018 consists of 14 video sequences of abdominal porcine procedures. Each video contains 300 frames with a resolution of $1280 \times 1024$, each annotated with 8 pixel-level categories. SurgDVC is an in-house dataset comprising 1003 high-resolution frames ($1920 \times 1080$) collected from real-world surgical procedures. The dataset was collaboratively annotated by three domain experts, who independently scored all segmentation results and reached consensus through discussion, ensuring highly reliable annotation quality. This dataset shares a subset of the anatomical categories defined in EndoVis2018, maintaining evaluation metric consistency while addressing domain-specific challenges. The dataset specifically captures surgical scenes under various complex visual conditions commonly encountered in clinical environments, including: low-light environments, presence of blood and surgical fluids, smoke from electrocautery procedures, and tissue occlusion—all representing challenging visual scenarios. Our experiments utilize the Mean of class-wise Intersection over Union (mIoU) and the Mean Dice coefficient (mDice) metrics to measure the performance of adaptation in semantic segmentation.

## 4.2 IMPLENTATION DETAILS

The base segmentation model is DeepLabv3+ (Chen et al., 2018a), which uses a pre-trained CLIP-ResNet-50 backbone (Radford et al., 2021). This model is thoroughly trained on the source domain following the configuration of (Fahes et al., 2022). In the target description generation process, we use LLaVA (Liu et al., 2023) as the VLM for text generation task. In Prompt-driven Local and Global Feature Stylization, the number of patches $m$ is 3, and the loss weights $\lambda_{local}$ and $\lambda_{global}$ are set to 0.1 and 1.0, respectively. In the fine-tuning stage (Stage 2), we begin with the source pre-trained model and fine-tune only the classifier head, following the configurations in (Fahes et al., 2022) for a fair comparison, and the loss weights $\lambda$ is set to 0.5. All models are tested on the original images without resizing, and further details are provided in the supplementary file.

## 4.3 COMPARISON WITH STATE-OF-THE-ART

We compare our approach with state-of-the-art baselines: source only model, CLIPstyler (Kwon & Ye, 2022) for zero-shot style transfer as well as zero-shot adaptation methods such as PØDA (Fahes et al., 2022), ULDA (Yang et al., 2024) and PiCaZo (Sikdar et al., 2025).

Table 1 reports results for adaptation from Cityscapes (clear weather) to ACDC (adverse weather). Our method achieves the highest average mIoU of 42.96, surpassing all prior approaches. This im-

Table 1: Performance comparison across multiple adaptation scenarios.*denotes results reproduced using official code.

| Scenarios | Source2Fog | | Source2Rain | | Source2Night | | Source2Snow | | Mean mIoU |
|---|---|---|---|---|---|---|---|---|---|
| Method | mIoU | mDice | mIoU | mDice | mIoU | mDice | mIoU | mDice | |
| Source-only | 49.98 | 63.91 | 38.20 | 50.78 | 18.31 | 28.01 | 39.28 | 49.64 | 36.44 |
| CLIPStyler*(Kwon & Ye, 2022) | 48.87 | 62.31 | 37.02 | 49.02 | 20.91 | 32.62 | 40.27 | 50.43 | 36.77 |
| PØDA*(Fahes et al., 2022) | 51.43 | 64.98 | 42.15 | 54.55 | 25.04 | 36.78 | 43.16 | 55.84 | 40.45 |
| ULDA*Yang et al. (2024) | 53.31 | 66.51 | 43.59 | 56.04 | 24.94 | 37.05 | 44.79 | 57.25 | 41.66 |
| PiCaZo(Sikdar et al., 2025) | 53.82 | - | 44.47 | - | 22.75 | - | 43.94 | - | 41.25 |
| FineDA | **54.12** | **67.21** | **45.23** | **58.12** | **25.82** | **37.31** | **46.67** | **59.31** | **42.96** |

provement highlights the effectiveness of our framework. The gain can be attributed to the Scene Graph-guided Chain-of-Thought module, which produces rich, image-specific prompts that capture scene semantics under challenging lighting and environmental conditions. In addition, the Prompt-Guided Local and Global Stylization enables fine-grained spatial adaptation, which is particularly beneficial for non-uniform shifts such as glare and shadows that are common in ACDC. The joint local-global normalization design provides greater robustness compared to methods relying only on global prompts. We further incorporate a generation step into our framework, which yields additional performance gains. Moreover, t-SNE visualizations in the appendix confirm that the generated features are distributed closer to real target features, offering further evidence of the effectiveness of our design.

**Cityscapes→GTA5**. As shown in Figure 4, our method achieves an average mIoU of 43.4, outperforming CLIPStyler (+4.7), PØDA (+2.3), and ULDA (+1.4). The GTA5 dataset introduces strong and spatially non-uniform domain shifts due to synthetic textures and inconsistent rendering. Our Scene Graph-guided Chain-of-Thought module provides more realistic and semantically accurate task descriptions, enabling precise class-level adaptation. Combined with local-global stylization, our method effectively handles uneven domain gaps and improves generalization to synthetic data. We give qualitative results in the Appendix.

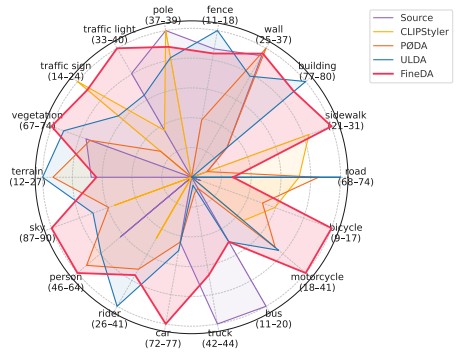

Figure 4: Performance comparison when trained on Cityscapes and tested on GTA5. The value is mIoU value.

**EndoVis2018→SurgDVC**. Table 2 presents the results on our in-house surgical dataset SurgDVC, which features strong spatial variation due to smoke, blood, and occlusion. These variations significantly impact the structure and details of the images. In this context, when confronted with unpredictable and rapidly changing visual artifacts, diffusion models may fail to precisely simulate these local shifts, resulting in image distortions or the loss of crucial details, which negatively impacts performance. Our method achieves an average mIoU of 21.61, outperforming PØDA by 2.62 and ULDA by 2.2. Additionally, our method also surpasses generation-based methods by 2.12, further demonstrating its robustness in handling complex challenges. These results highlight the capacity of our model to handle spatially non-uniform domain shifts, where purely global transformations fail. The patch-wise prompt-driven stylization allows the model to adapt regions affected by different visual artifacts (e.g., dark tools vs. bright background) independently, while still maintaining contextual consistency via the scene-level prompt.

## 4.4 ABLATION STUDY

To evaluate the effectiveness of our proposed method, we performed ablation studies. The framework consists of several components: Scene Graph-guided Chain-of-thought for Target Description Generation (SGCOT), Prompt-driven Local and Global Feature Stylization (PLGFS), and Feature Alignment Regularizer (FAR). Mean-mIoU is used as metrics, representing the average mIoU value across four scenarios from Cityscapes to ACDC.

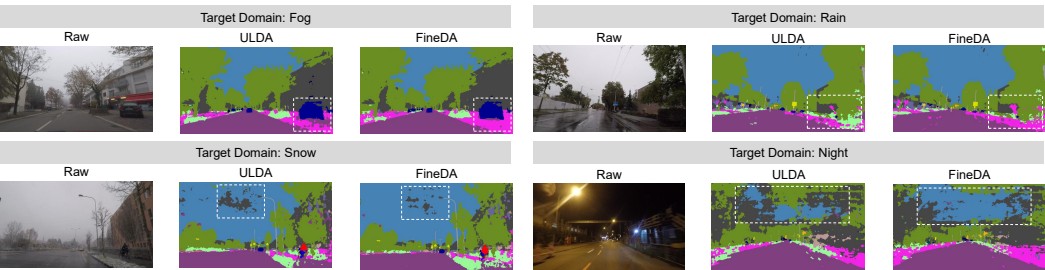

Figure 5: Qualitative comparison results on the task of Cityscapes → ACDC

Table 2: Performance comparison on the EndoVis2018→SurgDVC scenario.

| Method | mIoU | mDice |
|---|---|---|
| Source-only | 16.36 | 21.59 |
| CLIPStyler (Kwon & Ye, 2022) | 13.28 | 16.73 |
| PØDA (Fahes et al., 2022) | 18.99 | 24.79 |
| ULDA (Yang et al., 2024) | 19.41 | 25.62 |
| SDGPA (Luo et al., 2025) | 19.49 | 24.86 |
| FineDA | **21.61** | **28.75** |

Table 3: Evaluation on each component design. One-step represents using single-step prompts for target domain description generation.

| | One-step | SGCOT | PLGFS | FAR | Mean-mIoU |
|---|---|---|---|---|---|
| $Ex_1$ | | | | | 40.45 |
| $Ex_2$ | ✓ | | | | 41.13 |
| $Ex_3$ | | ✓ | | | 41.76 |
| $Ex_4$ | ✓ | | ✓ | | 42.13 |
| $Ex_5$ | | ✓ | ✓ | | 42.43 |
| $Ex_6$ | | ✓ | ✓ | ✓ | **42.96** |

**Effects of SGCOT and PLGFS**. Table 3 shows the baseline method in the first row. We also compared SGCOT with a one-step generation method to highlight its advantages. In $Ex_2$, we introduces One-step to align image embeddings and contextual target domain descriptions. This method results in improvements of 0.68% in mean-mIoU. In comparison, the $Ex_3$ introduces SGCOT, which results in improvements of 1.31% in mean-mIoU. $Ex_4$, based on $Ex_2$, introduces PLGFS, yielding a 1.00% improvement in mean-mIoU. Similarly, $Ex_5$, based on $Ex_3$, shows a 0.67% improvement with PLGFS integration. This signifies that incorporating Global and Local Style Transfer effectively addresses spatially non-uniform domain shifts and enhances contextual alignment. For a detailed examination of the impact of LGST, refer to the further ablation study provided in the Appendix.

**Effects of FAR**. Building on the $Ex_5$, the $Ex_6$ introduces Feature Alignment Regularizer. By integrating a mechanism that computes the similarity between image features and textual embeddings through the auxiliary loss, this signifies that FAR addresses discrepancies between simulated and real target domain features, ensuring better alignment during the fine-tuning phase. This results in a performance improve ment of 0.24% in mean-mIoU. This result highlights their complementary effects in the adaptation process.

This demonstrates that each component complements the others, collectively tackling the challenges of language-driven zero-shot domain adaptation. Due to the limitation of the space, we put training details, visualization of feature distributions, and key hyper-parameter choices, and training strategies in the Supplementary.

## 5 CONCLUSION

We presented FineDA, a novel framework for zero-shot domain adaptive semantic segmentation that addresses the core challenge of the simulated versus real feature gap. This gap arises from prompt-induced discrepancy and description mismatch, where coarse prompts fail to capture image-specific semantics and global normalization ignores spatially diverse domain shifts. FineDA introduces a scene graph guided chain-of-thought module for generating rich target descriptions tailored to each image, and a prompt guided local and global stylization module for fine-grained class-specific adaptation. A visual text feature alignment regularizer further enforces pixel-level semantic consistency. Experiments on public benchmarks and a challenging surgical dataset demonstrate the effectiveness and generalization of our approach.

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

# A APPENDIX

## A.1 THE USE OF LLM

We used a large language model (GPT-4) exclusively for language polishing and grammar checking during the writing of this paper. The underlying research, ideas, and findings are solely those of the author, who is fully responsible for the entire content of the manuscript.

## A.2 EXPERIMENTAL DETAILS

In SGCOT, FineDA leverages LLaVA to generate textual descriptions for the target domain, and then employs Qwen to simplify these descriptions to fewer than 77 tokens. Specifically, we employ a LLM to enhance each target domain description, ensuring that only the classes found in the corresponding ground truth segmentation mask are preserved, as shown in Fig. 6. This strategy helps reduce hallucination issues, since each source image may only include a subset of the total possible classes. In PLGFS, we employ the Stochastic Gradient Descent (SGD) optimizer, with a learning rate of 0.1 and a batch size of 8 over 100 iterations to train the PIN modules across all target domains. When fine-tuning the segmentation head of the target segmenter, we adopt the SGD optimizer with a learning rate of 0.01 and a batch size of 8 for 2,000 iterations. All experiments are performed using NVIDIA 4090 GPUs.

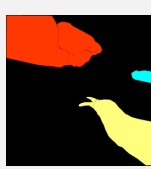

**Context Simplify : ⊟LLM**

**② Question：**

Simplify the description to less than 75 tokens. Make sure to include the class name [bipolar forceps, suction irrigator, curved scissors, background tissue] and the relationships of them. The description is **{VLM_CAPTION}**.

**🤖 Answer：**

The image shows a close-up view of a laparoscopic surgery, featuring a Bipolar Forceps on the left, holding tissue; a Grasper on the lower left, used to manipulate tissue; and Monopolar Curved Scissors near the center, positioned for cutting tissue. The background reveals soft tissue with a visible network of blood vessels.

Figure 6: Full prompt example of simplifying the text. We summarize generated captions with an LLM.

| Task | Method | road | sidewalk | building | wall | fence | pole | traffic light | traffic sign | vegetation | terrain | sky | person | rider | car | truck | bus | train | motorcycle | bicycle | mIoU |
|---|---|---|---|---|---|---|---|---|---|---|---|---|---|---|---|---|---|---|---|---|---|
| CS→GTA5 | Source | 68.7 | 22.7 | 78.8 | 36.8 | 17.3 | **39.7** | 39.3 | 14.8 | 72.6 | 22.5 | 87.3 | 57.5 | 26.1 | 74.3 | 44.6 | **20.5** | 0.0 | 18.3 | 10.4 | 39.6 |
| | CLIPStyler | 73.1 | 29.9 | 77.9 | 25.5 | 11.7 | 39.7 | 35.9 | **24.0** | 67.4 | 12.8 | 88.8 | **46.6** | 33.4 | 72.0 | 42.8 | 11.1 | 0.0 | 28.8 | 14.6 | 38.7 |
| | PØDA | 73.9 | 22.7 | 78.8 | **37.5** | 14.2 | 37.0 | 33.1 | 17.3 | 72.4 | 26.2 | 88.9 | 62.7 | 37.0 | 74.3 | 43.0 | 11.9 | 0.0 | 35.3 | 13.9 | 41.1 |
| | ULDA | **74.8** | 21.6 | **80.8** | 34.9 | **18.1** | 39.2 | 38.0 | 20.7 | 73.7 | **27.2** | 89.2 | 60.5 | **41.4** | 74.6 | 42.9 | 15.8 | 0.0 | 36.0 | 9.9 | 42.0 |
| | FineDA | 70.4 | **31.5** | 80.5 | 37.0 | 17.1 | 39.4 | **40.8** | 23.2 | **74.3** | 22.0 | **90.0** | **64.1** | 37.7 | **77.2** | 44.0 | 15.8 | 0.0 | **41.5** | **17.8** | **43.4** |

Table 4: Performance comparison between Cityscapes and GTA5. CS→GTA5 represents the adaptation task.

## A.3 CITYSCAPES→GTA5

The detailed class-wise results are reported in Table 4. Our approach attains mIoUs of 43.4, yielding relative gains of 3.8 compared with the original CLIP backbone. When set against CLIPStyler, the improvements become even more notable, with margins of 4.7. These findings confirm the effectiveness of our design in transferring CLIP between synthetic and real-world data.

## A.4 ADAPTATION TIME

As shown in Table 5, the adaptation time of our method is compared with PODA and ULDA across two tasks: adapting from Cityscapes (CS) to ACDC, and from EndoVis2018 to SurgDVC. While our method (FineDA) requires more adaptation time compared to PODA and ULDA, this is a direct result of the more sophisticated feature simulation and alignment processes used in our framework. These processes contribute to the significant performance gains we reported in the main text, particularly in terms of mIoU.

Table 5: Adaptation time on the adaptation stages.

| Method | Adaptation Time | |
|---|---|---|
| | CS→ACDC | EndoVis2018→SurgDVC |
| PODA | $2.97 \times 10^3$ | $1.41 \times 10^3$ |
| ULDA | $1.67 \times 10^4$ | $7.86 \times 10^3$ |
| SDGPA | $1.84 \times 10^4$ | $8.42 \times 10^3$ |
| *FineDA* | $\mathbf{1.93 \times 10^4}$ | $\mathbf{9.24 \times 10^3}$ |

## A.5 FEATURE VISUALIZATION

To evaluate the effectiveness of the proposed method in simulating the stylistic features of target domain images, we performed t-SNE visualization analysis on the style distribution of the generated

features. Fig. 7 presents the visualization results of feature distributions from our method, alongside those of PØDA, ULDA, and the actual target domain images. The comparative visualization clearly shows that the features generated by our method are distributed much closer to the actual target domain features than those of other methods. This evident reduction in the distribution gap confirms the superiority of our approach in style feature simulation.

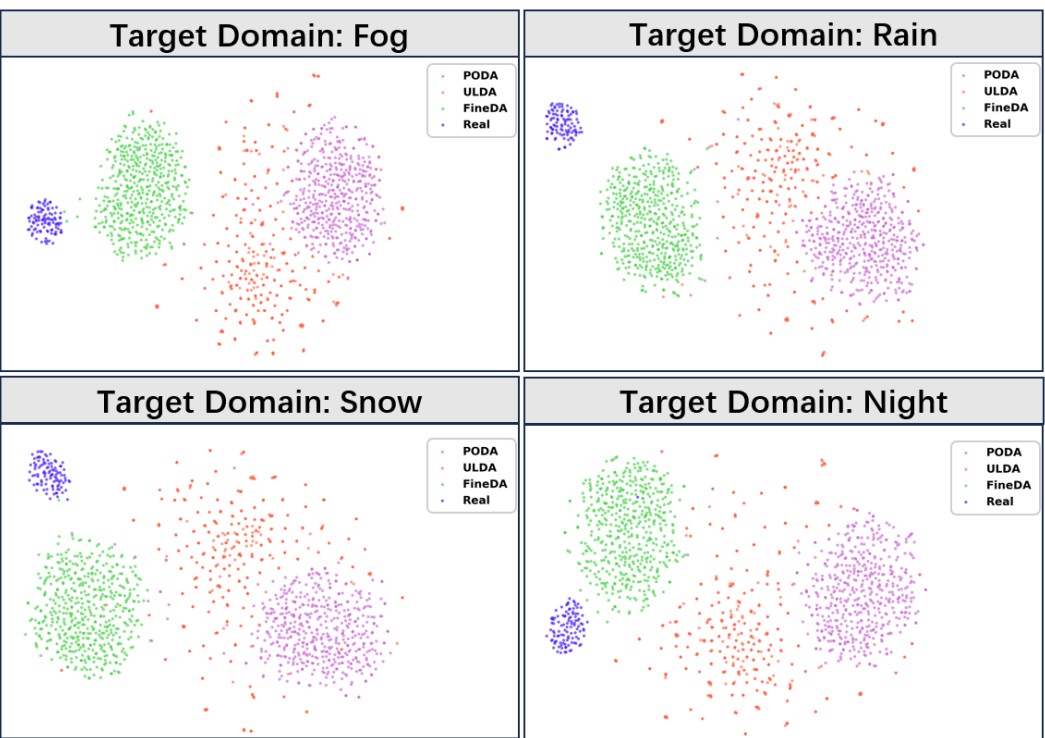

Figure 7: The visualization results of the simulated target domain style obtained through each method.

### A.6 OPTIMIZATION STEPS

In all our experiments, 100 iterations of optimization are performed for each batch of source features. We show in Fig. 8a the effect of the total number of iterations. We see an inflection point at the 100th iteration. Using few iterations is not sufficient for style alignment. Above 100, we also observe a performance drop.

### A.7 PATCH NUMBERS

To investigate the effect of patch number on experimental results, we designed comparative experiments with different numbers of patches. By increasing the number of patches, we can optimize each local region more finely, thereby improving the precision of style alignment. However, too many patches may lead to increased computational cost and instability in the optimization process. Therefore, it is necessary to balance the number of patches with optimization efficiency. As shown in Fig. 8b, we conducted experiments with different patch numbers and presented the performance of each configuration to verify the importance of patch number in the optimization process.

### A.8 IMPACT ANALYSIS OF DIFFERENT LARGE LANGUAGE MODELS

In this study, we explored the use of various large language models (LLMs), including both open-source and proprietary ones, to generate target-domain descriptions for guiding feature stylization. Experimental results indicate that, although the generated texts differ in detail and style, the final segmentation performance remains largely consistent. We attribute this observation to the following reasons:

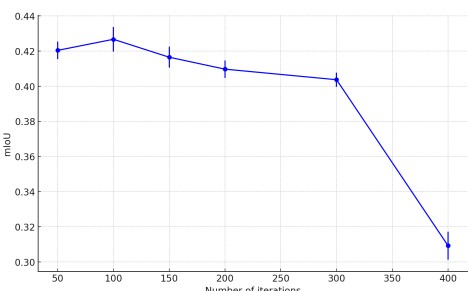 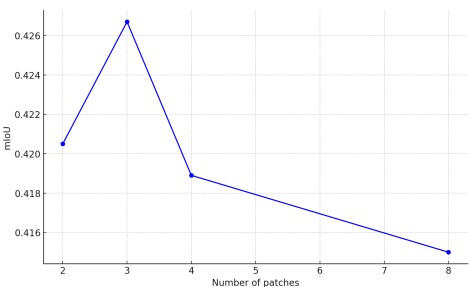

(a) Effect of the number of optimization iterations. Mean-mIoU is used as metric, representing the average mIoU value across four scenarios from Cityscapes to ACDC.

(b) Effect of the number of patches. Mean-mIoU is used as metric, representing the average mIoU value across four scenarios from Cityscapes to ACDC.

Figure 8: Comparison of optimization iterations (a) and patch numbers (b) in our framework.

First, the baseline model employs ResNet-50 as the visual encoder of CLIP, which is kept frozen during training. This setup inherently limits the model's sensitivity to variations in textual descriptions, leading to similar adaptation performance regardless of the specific LLM used.

Second, the CLIP text encoder is restricted to an input length of 77 tokens. As a result, overly long or detailed descriptions are truncated during encoding. Consequently, even if different LLMs produce semantically diverse outputs, their encoded vector representations exhibit only minor differences, making it difficult for such variations to significantly influence stylization outcomes.

Finally, while LLMs may occasionally generate inaccurate or biased descriptions, our method incorporates a local class-wise text alignment mechanism. This design provides additional robustness during training, enabling the model to effectively disregard erroneous or noisy text and thereby enhancing fault tolerance.

In summary, these factors collectively explain why our method achieves stable segmentation performance across target-domain descriptions generated by different LLMs.

