# OpenReview forum: "Fine-Grained Prompt-Driven Stylization with Context-Aware Reasoning for Zero-Shot Domain Adaptation"
_ICLR.cc/2026/Conference — ICLR 2026 Conference Desk Rejected Submission_

### Official Review · Reviewer_zVFA · 2025-10-26

**Soundness:** 2
**Presentation:** 2
**Contribution:** 3
**Rating:** 4
**Confidence:** 3

**Summary:**

The paper tackles zero-shot domain adaptive semantic segmentation (ZSDA) without target images. It identifies two key issues in prompt-driven stylization: description mismatch and prompt-induced discrepancy. FineDA proposes: (1) a Scene Graph–guided Chain-of-Thought (SGCOT) to generate image-specific, context-rich target descriptions; (2) a Prompt-driven Local and Global Feature Stylization (PLGFS) that performs patch-wise, class-aware PIN alongside a global prompt; (3) a Visual-Text Feature Alignment Regularizer (FAR) akin to DenseCLIP to improve pixel-level alignment during fine-tuning.

**Strengths:**

1. This paper has clear problem statement and motivation, i.e., addressing the gap between simulated and real-world characterizations in ZSDA with concrete failure modes.
2. The proposed SGCOT for prompt generation and PLGFS for joint local-global PIN are reasonable, complementary extensions to prior prompt-driven stylization.

**Weaknesses:**

1. SGCOT uses the ground truth mask to build the scene graph and select the class names for local cues, which is valid in ZSDA (the labels are present in the source). However, it should be clarified that the target label or image is never used for cue generation, and any VLM/LLM conditioning only sees the source image and source mask. Safeguards to prevent indirect use of target statistics should be explicitly stated.
2. PLGFS instantiates per-patch PIN with class-specific prompts. With m>1, does each patch get a separate PIN instance or a shared predictor conditioned on the prompt? Training m independent PINs per image can be memory-heavy and brittle.
3. The scene graph creation depends on VLM outputs. How often do object hallucinations occur, and how effectively does the LLM filter mitigate them?
4. FAR defines a cost map and uses cross-entropy on cosine similarities. Cosine scores can be negative and unnormalized and detail the softmax step to ensure valid probabilities.
5. You are suggested to discussing failure modes in extreme night scenes with sparse class presence, rare classes without reliable textual synonyms, or out-of-vocabulary classes for CLIP text encoder.
6. Fix typos and consistency (e.g., “Implentation” → “Implementation”, consistent use of ResNet-50 naming and ensure consistent λ notation between sections).

**Questions:**

See Weaknesses

---

### Official Review · Reviewer_z5Yo · 2025-10-31

**Soundness:** 2
**Presentation:** 3
**Contribution:** 2
**Rating:** 2
**Confidence:** 4

**Summary:**

This paper addresses the challenge of Zero-Shot Domain Adaptive Semantic Segmentation, where models must generalize to unseen target domains without accessing target domain data during training. The authors propose a framework that tackles two key problems in existing methods: description mismatch, where coarse textual prompts fail to capture scene-specific semantics, and prompt-induced discrepancy, where uniform global stylization cannot handle spatially non-uniform domain shifts. The authors report state-of-the-art results in terms of mIoU and mDice, showing significant improvements over prior methods like PØDA, ULDA, and PiCaZo.

**Strengths:**

1. The paper tackles the practical problem of zero-shot domain adaptation for semantic segmentation, particularly useful in safety-critical applications like medical imaging and autonomous driving. The introduction of a surgical dataset with complex visual artifacts adds value to the evaluation.

2. The paper includes detailed ablation studies to evaluate the contributions of individual components (SGCOT, PLGFS, FAR), showing how each module improves performance. This demonstrates the complementary nature of the components.

3. The use of scene graphs for generating semantically rich, image-specific prompts is a novel and promising approach. By incorporating object attributes and relationships, the framework creates more contextually relevant descriptions, which improve stylization and alignment with the target domain.

**Weaknesses:**

1. While the SGCOT and PLGFS modules are novel in their combination, they build upon existing techniques rather than introducing fundamentally new concepts. For example, Prompt-Instance Normalization (PIN) is extended to a patch-wise setting, and chain-of-thought reasoning with scene graphs has been explored in related works. The overall contribution feels more like a refinement of existing ideas rather than a breakthrough.

2. The paper does not provide sufficient justification for several design choices, such as the specific number of patches, the weighting of local versus global losses, or the use of a two-step chain-of-thought reasoning approach. While these choices are empirically validated, their theoretical basis is unclear.

3. While quantitative results are strong, the qualitative analysis is limited. More examples of segmentation outputs, particularly in challenging regions (e.g., occlusions, glare), would help illustrate the method’s strengths.

4. The patch-wise processing in PLGFS and the two-step chain-of-thought reasoning in SGCOT introduce significant computational costs, making the method less practical for real-time or large-scale applications. As shown in Table 5, FineDA has a longer adaptation time compared to PØDA and ULDA, which raises concerns about its scalability in resource-constrained settings.

**Questions:**

Please refer to the weaknesses above.

---

### Official Review · Reviewer_uSaS · 2025-10-31

**Soundness:** 3
**Presentation:** 2
**Contribution:** 2
**Rating:** 4
**Confidence:** 3

**Summary:**

This paper proposes **FineDA**, a zero-shot domain adaptation framework for semantic segmentation that combines context-aware prompt reasoning with fine-grained local–global stylization.  The method introduces three components: a Scene Graph-Guided Chain-of-Thought (**SGCOT**) module for structured prompt generation, a Prompt-Driven Local and Global Feature Stylization (**PLGFS**) block for patch-level alignment, and a Visual–Text Feature Alignment Regularizer (**FAR**) enforcing pixel-level semantic consistency with CLIP embeddings.  **FineDA** achieves consistent gains over **PØDA**, **ULDA**, and other baselines on *Cityscapes→ACDC*, *Cityscapes→GTA5*, and a new surgical dataset (*SurgDVC*).

**Strengths:**

1. **Well-motivated problem** – The work addresses the lack of fine-grained semantic reasoning in prompt-based ZSDA, which is timely given the increasing interest in vision–language alignment and zero-shot generalization.

2. **Comprehensive experiments** – Evaluations span multiple datasets and include a medical setting, highlighting cross-domain robustness.

3. **Reasonably clear methodology** – The modular design (SGCOT, PLGFS, FAR) is well-defined and follows a consistent training objective.

4. **Empirical improvement** – Reported quantitative results are strong and consistent across benchmarks.

**Weaknesses:**

1.  **Weak justification of design choices** – The benefit of using **scene graph–guided reasoning** and **local–global stylization** on top of CLIP is unclear. CLIP’s text encoder can only process 77 tokens, making such complex reasoning potentially ineffective. For relatively simple VLMs like CLIP, well-structured but concise prompts may already be optimal, and it is uncertain whether the proposed modules genuinely improve alignment.
2. **Limited theoretical depth** – The core contributions are primarily architectural and empirical. The paper lacks deeper theoretical justification or analysis of why fine-grained stylization leads to measurable alignment improvement beyond empirical observation.

3. **Reliance on ground-truth masks** – The construction of scene graphs assumes access to segmentation masks, which weakens the “true zero-shot” claim. Without a mask-free or pseudo-label alternative, the method’s general applicability is constrained.

4. **Computational complexity** – The adaptation process is significantly more expensive (≈6× runtime of PØDA). No clear mitigation strategy or runtime–accuracy trade-off analysis is provided.

5. **Incremental innovation** – Each individual component (PIN-based normalization, CLIP alignment, CoT reasoning) builds directly upon existing techniques. The novelty primarily lies in integration rather than fundamentally new insight.

6. **Clarity and presentation** – Figures are visually dense, and some key concepts (e.g., local/global feature separation) could benefit from clearer visualization. The exposition in Section 3 is heavy and difficult to follow for general readers.

**Questions:**

1. **How can the proposed framework achieve a truly mask-free scene graph construction?**
   The current design seems to rely on segmentation masks to generate scene graphs, which contradicts the zero-shot assumption.
   Could the authors explore unsupervised detection or attention-based parsing to build scene graphs without labeled masks?

2. **What concrete evidence supports that the reasoning and stylization modules improve CLIP feature alignment?**
   Given CLIP’s limited 77-token text encoder, complex reasoning may not yield proportional gains.
   Have the authors conducted diagnostics (e.g., feature-space distance metrics or t-SNE visualization) to show measurable improvements in domain alignment?

3. **How can the presentation and formulation be simplified for clarity?**
   The mathematical sections are dense, and figures are visually heavy.
   Can the authors improve figure readability and streamline notation to make the core ideas more accessible?

4. **Can FineDA generalize beyond segmentation?**
   The current validation focuses solely on semantic segmentation.
   Have the authors tested or considered extensions to object detection, depth estimation, or broader multimodal reasoning tasks?

---

### Note · Program_Chairs · 2026-01-17
**Submission Desk Rejected by Program Chairs**

The following references in this submission do not refer to real documents and/or have major errors in bibliographic information:

 Yi-Hsuan Tsai, Wei-Chih Hung, Samuel Schulter, Ki hyuk Sohn, Ming-Hsuan Yang, and Manmohan Chandraker. Title of the paper. In Proceedings of the Conference Name, 2023.